# Changes in the Human Gut Microbiome during Dietary Supplementation with Modified Rice Bran Arabinoxylan Compound

**DOI:** 10.3390/molecules28145400

**Published:** 2023-07-14

**Authors:** Emily Schupfer, Soo Liang Ooi, Thomas C. Jeffries, Shaoyu Wang, Peter S. Micalos, Sok Cheon Pak

**Affiliations:** 1School of Dentistry and Medical Sciences, Charles Sturt University, Bathurst, NSW 2795, Australia; eschupfer@csu.edu.au (E.S.); sooi@csu.edu.au (S.L.O.); 2School of Science, Western Sydney University, Penrith, NSW 2751, Australia; t.jeffries@westernsydney.edu.au; 3School of Dentistry and Medical Sciences, Charles Sturt University, Orange, NSW 2800, Australia; 4Ageing Well Research Group, Charles Sturt University, Orange, NSW 2800, Australia; 5School of Dentistry and Medical Sciences, Charles Sturt University, Port Macquarie, NSW 2444, Australia; pmicalos@csu.edu.au

**Keywords:** RBAC, Biobran/MGN-3, microbiome, prebiotic, functional food, diet, Australian Recommended Food Score

## Abstract

This study investigated the effects of a modified rice bran arabinoxylan compound (RBAC) as a dietary supplement on the gut microbiota of healthy adults. Ten volunteers supplemented their diet with 1 g of RBAC for six weeks and 3 g of RBAC for another six weeks, with a three-week washout period. Faecal samples were collected every 3 weeks over 21 weeks. Microbiota from faecal samples were profiled using 16S rRNA sequencing. Assessment of alpha and beta microbiota diversity was performed using the QIIME2 platform. The results revealed that alpha and beta diversity were not associated with the experimental phase, interventional period, RBAC dosage, or time. However, the statistical significance of the participant was detected in alpha (*p* < 0.002) and beta (weighted unifrac, *p* = 0.001) diversity. Explanatory factors, including diet and lifestyle, were significantly associated with alpha (*p* < 0.05) and beta (*p* < 0.01) diversity. The individual beta diversity of six participants significantly changed (*p* < 0.05) during the interventional period. Seven participants showed statistically significant taxonomic changes (ANCOM W ≥ 5). These results classified four participants as responders to RBAC supplementation, with a further two participants as likely responders. In conclusion, the gut microbiome is highly individualised and modulated by RBAC as a dietary supplement, dependent on lifestyle and dietary intake.

## 1. Introduction

Communities of microorganisms reside within humans, with the gut being the most densely colonised site [1]. Microbial communities that cohabit the intestinal tract, known as gut microbiota, can be beneficial, neutral, or detrimental to the host. Gut microbiota can impact human health by mediating physiological homeostasis through immune function, digestion, vitamin synthesis, and pathogen colonisation [2]. The effects are exerted through interfaces between gut microbiota and intestinal epithelial cells, the immune system, and dietary intake [2,3,4]. Gut microbiota diversity and abundance is associated with health through processes such as reduced inflammation and disorders linked to inflammatory events [3].

Microbial diversity is required for healthy gut function, with a loss of diversity associated with an increased risk of disease [5]. In healthy adults, the gut microbiota comprises eight dominant phyla, including Firmicutes, Bacteroidetes, Actinobacteria, Proteobacteria, Fusobacteria, and Verrucomicrobia. The Firmicutes and Bacteroidetes together represent 90% of gut microbiota [6,7]. Bacteria in these phyla perform diverse roles in regulating host health [7]. *Bifidobacterium* species comprise only 2% of the total gut microbiota but play a significant role in the breakdown of complex carbohydrates [8,9], protecting the host against pathogens through competitive exclusion, modulating the immune system, and providing vitamins and other nutrients for the host [10]. The genus *Lactobacillus* includes species that produce short-chain fatty acids (SCFAs) and regulate intestinal transit [11,12]. The production of SCFAs by these bacteria can improve insulin sensitivity and protect against diet-induced obesity [9]. Other bacteria of importance to human health include *Roseburia*, *Faecalibacterium*, *Ruminococcus*, and *Bacteroides* [13,14,15].

The presence of microbes and their interaction with the host and other gut microbes in the gastrointestinal tract can lead to adverse health outcomes [1]. Metabolic products from gut microbiota have been linked to an increased risk of major adverse cardiovascular events [16] and type 2 diabetes [17]. Supporting an optimal composition of gut microbiota is vital for human health [18]. The gut microbiota in healthy individuals includes shared common microbiota, with variations across age, ethnicity, and socioeconomic status [19,20,21,22,23]. Although relatively stable throughout adulthood, variations in the common microbial genera are observed, with major shifts identified in older adults [23,24,25,26]. The composition of the gut microbiome is also affected by lifestyle [27,28]. Cigarette smoking and alcohol consumption impact the gut microbiome with changes identified with the cessation of smoking, and heavy alcohol consumption [27,28].

Diet has also been shown to be a significant factor affecting gut microbial composition [18,29]. Functional foods aim to improve health and wellbeing and have been shown to interact with gut microbiota [30,31,32]. Prebiotics are non-digestible dietary ingredients that stimulate the growth of microbes in the gut after fermentation [33]. Recent research identifies prebiotics that enhance microbial diversity to promote health and defend against the dysbiosis of gut microbiota [4,5]. Arabinoxylans (AXs) are the non-digestible fibre of cereal grains, including wheat, rice, rye, maize, and sorghum [9]. The structures of AXs have been shown to influence substrate fermentation and degradation by gut microbiota [34]. Francios et al. [35] and Kjolbaek et al. [36] showed that supplementation with wheat AX/AX oligosaccharides in humans led to changes in the microbiota profile, including increased beneficial bacteria, such as *Bifidobacterium*. Lachnospiraceae have also been shown to be increased in AX-fed mice [37]. The extent and mechanism of the effects of AXs on gut microbiota differs due to variations in the chemical structure and molecular weight [38]. Recent in vivo research has focused on linear AXs due to their bioavailability and simple structures [9,39].

Rice bran arabinoxylan compound (RBAC), known as BioBran, MGN-3, and Ribraxx, is a modified AX compound [40]. RBAC consists of a water-soluble hemicellulose-β fraction (degree of polymerisation approximately 200), partially decomposed by enzymes extracted from a cultured medium of *Lentinula edodes* (shiitake mushrooms) [40]. The production of RBAC typically involves the preparation of rice bran in a growth medium with sterilisation, followed by bioconversion or fermentation with *L. edodes* enzyme for a set time before the extraction, purification, and drying of the compounds into powder form [31]. The active ingredients of RBAC comprise heteropolysaccharides with a primary structure of arabinoxylan with a xylose in its main chain and an arabinose polymer side chain as shown in Figure 1 [41,42].

RBAC is safe to consume with a median lethal dose (LD_50_) above 36 g/kg body weight, and the no-observed-adverse-effect level is above 200 mg/kg/day [43]. The compound also tested negative in a reverse mutagenicity array and did not elicit any allergic response in a controlled antigenicity test [43]. A safety study conducted with 24 healthy volunteers given RBAC at different concentrations of 15, 30, and 45 mg/kg body weight daily for one month also reported no abnormalities detected in blood chemistry analysis, including liver enzymes, as compared to the baseline [44]. RBAC also has an excellent safety record. In the systematic review of RBAC, no adverse events at the typical dosage of 1–3 g/day were reported in any of the included clinical trials (total 11) or clinical case reports (total 14) [45]. Hence, RBAC is considered safe to consume with no known side effect at the typical dosages used in research and clinical settings.

RBAC has been widely used as a dietary supplement, most notable for its immunomodulating property [45]. However, knowledge of the effect of RBAC on gut microbiota, particularly in healthy adults, is lacking. Therefore, the purpose of this basic science study was to explore possible changes in gut microbiota in response to dietary RBAC supplementation in healthy adults. Additional observational analysis was also performed to assess the abundance of the beneficial bacteria *Bifidobacterium*, *Lactobacillus*, *Roseburia*, *Ruminococcaceae*, and *Faecalibacterium*.

## 2. Results

### 2.1. Participant Characteristics and Dietary Intake

Characteristics of the study participants and their dietary intake from the screening survey and Australian Eating Survey® (AES) are summarised in Table 1. The participant group consisted of balanced numbers of males and females (sex ratio 1:1). Ages ranged from 22 to 56 years (n = 10, average age of 30.6 years), with most participants (n = 8) being 30 years or younger (average age of 26.5 years), and two participants above 30 years (average age of 46.5 years). Nine participants did not smoke cigarettes, and six participants consumed alcohol. The dietary analysis showed that most participants did not follow a particular dietary type and were considered omnivores; however, two participants were either vegan or pescatarian. The Australian Recommended Food Score (ARFS) was calculated as an average from AES results obtained before intervention commencement and at the completion of the study. For the sampling, 80 faecal samples were collected, with 3 faecal samples lost in transit (P05 timepoints 3 and 4, and P09 timepoint 4). A total of 77 samples were analysed for this study.

### 2.2. DNA Quality

There were 14,883,521 high-quality 16S rRNA sequences obtained with a mean of 173,579 forward and reverse reads per sample from 77 samples.

### 2.3. Phylogenetic Taxonomy of the Gut Microbiota

The gut microbiota taxonomy for all samples from each participant are visualised at the phylum level using taxa bar plots (Figure 2). This graph shows that Bacteroidetes (green) and Firmicutes (purple) are the two dominant phyla across all samples. Furthermore, the plot shows that the taxa of gut microbiota in participants vary from each other. However, the ratio of Firmicutes/Bacteroidetes fluctuates over the time points and was not statistically significant.

### 2.4. Associations of Alpha Diversity of Gut Microbiota with Explanatory Factors

The rarefaction plot generated using Shannon’s index [46] shows a maximum depth of 47,647 based on the median frequency value from the frequency per sample results. This is sufficient for analysis, as a levelling out on the *y*-axis was observed at this depth, indicating that additional sequences beyond this depth would unlikely result in additional observed features.

The association of alpha diversity metrics with explanatory factors using Kruskal–Wallis analysis was performed on sequences pooled according to explanatory factors (Table 2). These factors are participant, sex, age group, cigarette smoking, alcohol consumption, and ARFS group. The association of alpha diversity was also determined for the time point (1–8), experimental phase (baseline, low dose, washout, high dose, and post-intervention), RBAC dosage (0 g/day, 1 g/day, and 3 g/day), and interventional period (baseline to experimental period including washout and post-intervention). Participant was the only factor that showed a statistically significant effect on alpha diversity in Shannon’s evenness (*p* = 9.66 × 10${}^{-8}$) and Faith’s PD (*p* = 0.002). Statistically significant changes in Shannon’s evenness were shown for alcohol consumption (*p* = 7.92 × 10${}^{-6}$, cigarette smoking (*p* = 0.008), sex (*p* = 0.003), and ARFS group (*p* = 0.032). There were no significant changes in alpha diversity across age group, time point, experimental phase, RBAC dosage, and interventional period.

### 2.5. Associations of Beta Diversity of Gut Microbiota with Explanatory Factors

Associations of beta diversity with explanatory factors were determined using PERMANOVA (Table 3). Participant sequence data were grouped according to the explanatory factor being tested using the same technique as alpha diversity analysis. Statistically significant differences in beta diversities were found across participant (*p* = 0.001), sex (*p* = 0.001), alcohol consumption (*p* = 0.001), ARFS group (*p* = 0.001), and age group (*p* ≤ 0.013). Cigarette smoking was significantly associated with all matrices (*p* = 0.001), except weighted unifrac (*p* = 0.1). The analysis for time point, experimental phase, RBAC dosage, and interventional period did not show statistically significant associations with any beta diversity measurements.

An EMPeror plot of Bray–Curtis beta diversity showing clustering according to individual participants is demonstrated in Figure 3. These association analyses show that participant is the major explanatory factor associated with both alpha and beta diversity patterns. This finding suggests that individuality is a critical determinant of microbial diversity.

These findings suggest the requirement to analyse beta diversity for each participant to determine whether there are statistically significant longitudinal changes from the baseline across the experimental period. Individual beta diversity for each participant across samples (1–8) was calculated using weighted unifrac and Bray–Curtis methods (Table 4).

The explanatory factors used for this analysis were RBAC dosage, experimental phase, and interventional period. In P02, the interventional period was significantly associated with weighted unifrac (*p* = 0.01) and Bray–Curtis (*p* = 0.012) beta diversity methods. In P03, the experimental phase and interventional period were significantly associated with both methods (*p* ≤ 0.052). P04 observed associated changes in RBAC dosage for both methods (*p* = 0.035 and *p* = 0.018, respectively), as well as the experimental phase with Bray–Curtis (*p* = 0.044). In P07, the experimental phase (*p* = 0.025) and interventional period (*p* = 0.025) were significantly associated with Bray–Curtis. In P08, the RBAC dosage was significantly associated with both methods (*p* < 0.05), and the experimental phase and interventional period were associated with Bray–Curtis only. P09 observed that the dosage and experimental phase are significantly associated (*p* < 0.05) with both methods. It is further noted that P05 and P10 showed marginally significant values in the Bray–Curtis method for the experimental phase.

The longitudinal findings are illustrated using the weighted unifrac beta diversity distance matrices across the eight time points in the EMPeror plots for the selected participants (Figure 4). For P03, all time points after baseline (red points) show a significant change. The graph for P04 illustrates an increase from the baseline in the low dose interventional period (orange points) but no change in the high dose (blue points) or washout (green points) periods. P01 shows no change across any time points from the baseline. Please refer to Appendix A of the Appendix A for the plots for all participants.

### 2.6. Detection of the Association of Microbial Diversity with Interactions of Explanatory Factors

Further investigation was carried out to determine whether multiple variables could explain the variation in participants’ gut microbiota and reduce false positives in the beta diversity analysis. The calculated unifrac distance matrix was used to perform an Adonis PERMANOVA test on participant and other explanatory factors, including RBAC dosage, experimental phase, time point, and interventional period. The results showed that the variable participant (R^2^ = 0.796, *p* = 0.001) explains approximately 80% of variance in the beta diversity. Regression analysis with linear models showed a statistical significance for the experimental phase (R^2^ = 0.019, *p* = 0.026) and intervention (R^2^ = 0.012, *p* = 0.011). Hence, after the participants’ variation is accounted for, the experimental and interventional periods explain approximately 3.1% of the variance observed in beta diversity with the remaining 18% of variance being unaccounted for. Hence, the potential effects of the experimental intervention cannot be completely dismissed.

### 2.7. Significant Taxonomic Changes from Baseline to Interventional Period

Taxonomic changes from baseline to the interventional period were statistically assessed using ANCOM. Data for all statistically significant changes with a W value of 5 or above are listed in Table 5. Associated changes were compared from the baseline to the total interventional period unless otherwise specified. These results show statistically significant taxonomic changes (from baseline) detected down to a species level for 7 participants—P01, P02, P03, P07, P08, P09 and P10.

In P01, a marked increase in the *Hungatella* genus with a W value of 40 was revealed and fewer marked changes across other genera and families were observed. In P02, decreases in *Anaerococcus*, *Corynebacterium*, and *Finegoldia* genera were detected with W values ranging between 48 and 64. Large declines in *Dialister* (W = 94) and *Gastranaerophilales* (W = 86) were observed in P03. A large decrease in *Erysipelatoclostridiaceae* (W = 102) was displayed in P07. In P08, there was a decrease in the *Eubacterium siraeum* group (W = 35) with lower significant changes in other bacterial genera. P09 revealed large increases in multiple bacterial genera, including *Eubacterium siraeum*, (W = 38). This participant also showed smaller increases in *Eubacterium hallii* group and *Stomatobaculum* (W = 5 for each) and further smaller increases in *Anaerococcus*, *Ruminococcus*, *Megasphaera*, *Solobacterium* (W = 7–8) in either low or high RBAC doses. P10 showed a large decrease in *Prevotella* (W = 20) and smaller changes across other groups. Some changes were only detected during the washout stage as observed in P01.

## 3. Discussion

This basic scientific study aimed to explore possible changes in the gut microbial composition in response to dietary RBAC supplementation in healthy adults. Our results revealed that alpha and beta diversity were not associated with the experimental phase, interventional period, RBAC dosage, or time point. This suggests that the dietary supplementation did not significantly moderate the gut microbiota composition. However, the absence of the supplementation effect on the gut microbiota composition could be affected by a couple of factors. Firstly, the unknown species are not accounted for in alpha diversity measurements. This indicates that the true richness and evenness of the microbial environment was not fully examined in the present study. Consequently, the results on alpha diversity may be misrepresented [47]. Secondly, the finding that participant is a significant determinant for alpha and beta diversity suggests that the individual microbiome requires separate assessment alongside the explanatory factors during RBAC supplementation on gut microbiota.

The present research findings revealed differences in the alpha and beta diversity of the gut microbiota within individual participants. The gut microbial composition presents a profile that is distinct for the individual, with moderating factors such as genetics, environment, and lifestyle [48]. Previous research on twins indicated heritable common gut microbiota, with cohabitating twins sharing the highest number of strains [49]. However, differences in the microbiota between twins were also shown in previous studies, indicating that non-genetic factors are also at play [48,49]. Since genetics and environment are known factors influencing gut microbiota composition, it was not unexpected that the microbial composition between participants significantly differed in the present study. Compounding these factors is the individual lifestyle, which further impacts microbiota diversity among participants [50].

The present results also revealed significant association between alpha (Shannon’s evenness) and beta diversity with alcohol consumption, cigarette smoking, and diet (Australian Recommended Food Score, ARFS). Alcohol consumption has been shown to affect gut microbiota and has been linked to substantial losses in diversity, including reductions in Bacteroidetes and Firmicutes [51]. Alcohol consumption also correlates with decreased connectivity of the microbial network and subsequent alteration of gut microbiome composition [28]. Cigarette smoking causes taxonomic changes in the gut microbiota, specifically decreased Bacteroidetes and increased Firmicutes, due to nicotine exposure [52]. Moreover, after smoking cessation, increased alpha diversity of gut microbiota was observed [27]. Due to the the toxic effects on gut mucosa, cigarette smoking has also been linked to gut microbial dysbiosis [52]. The present study identified lifestyle, including diet and subsequent ARFS, alcohol consumption, and cigarette smoking, as important factors that impact gut microbiota in healthy adults.

Five of the six participants in the present study with the highest ARFS scores revealed significant changes in beta diversity. Although there is limited research on how the ARFS is associated with the gut microbiome, Aslam et al. [53] reported that higher ARFS correlated with the consumption of a more diverse diet, with differences shown in the beta diversity compared to limited diets with lower ARFS. Hence, the association of ARFS with changes in beta diversity may be explained by the differences between the consumption of core foods (e.g., grains and water) and discretionary foods (e.g., fried products and packaged sweets) in the diet [54]. The increased consumption of core foods and low discretionary foods also shows a higher proportion of complex carbohydrates, which benefits gut microbiota. This may explain the significant changes observed in the beta diversity for participants with a higher ARFS in the present study [54].

Age group as an explanatory factor was revealed to be associated with significant changes in beta diversity. Considerable shifts in the microbiome were previously shown during infancy, puberty, to the later stage of life (>75 years of age) [21]. The stability of the microbiome is reached during adulthood (18–25 years of age) and remains relatively constant until approximately 75 years of age when a loss of diversity occurs [21]. However, Odamaki et al. [23] observed significant differences between adult clusters at the ages of 33 (cluster 1) and 42 (cluster 2), with higher levels of Bacteroidetes observed in the younger age group (cluster 1). This is consistent with the significant differences in beta diversity observed in the present study for participants ≤30 years and >30 years of age.

Associations between sex, and alpha and beta diversity were observed. Sex is a known modulator of the gut microbiome through the actions of sex hormones [55]. This is supported by recent studies showing that sex is a prime contributor to microbial diversity with increased alpha diversity observed in females compared to males [56,57]. Furthermore, sex has also been linked to responses of the gut microbiota to diet, anti-microbial effects, and obesity [55,58]. The present findings affirm that age and sex significantly influence the gut microbiota composition within healthy adults.

Dietary RBAC supplementation was shown to change the abundance of beneficial bacteria, including *Lactobacillus*, *Roseburia*, and *Ruminococcaceae*, although these changes were not observed in all participants. Some participants exhibited trends in taxonomic changes during the interventional period. For example, participant 9 (P09) showed increases across multiple genera. These beneficial bacteria play a vital role in gut health through the production of SCFAs as essential energy sources for colonic enterocytes [12,14], the provision of vitamins [10], and the possession of anti-inflammatory properties [13]. This finding suggests that dietary RBAC supplementation may be used as a prebiotic for regulating these beneficial bacteria.

The increases in beneficial bacteria observed in some participants may also be associated with the ARFS. Diets with less diversity have been associated with lower bacterial variety [59]. Of the four participants who exhibited taxonomic changes in beneficial bacteria, three had the lowest ARFS. Hence, this finding suggests a potential association between RBAC supplementation responses and ARFS. This indicates that RBAC supplementation may influence the growth of the beneficial gut microbiota of people with diets that are lower in core foods and higher in discretionary foods. In addition to increased beneficial bacteria, there were also reductions in bacteria known to be opportunistic pathogens, including *Anaerococcus* and *Corynebacterium* [60,61]. However, these taxonomic changes are difficult to interpret due to the difference in gut microbiota between individuals and how rapidly the gut microbiome can change in response to daily diet and lifestyle factors [50].

In six participants, significant changes in beta diversity were associated with RBAC dosage, experimental phase, and/or interventional period. Previous research has indicated that the composition and diversity of the gut microbiome can determine whether participants respond to interventions, including dietary modification for health-related purposes [62]. This is further extended to other interventions, such as treatments for cervical cancer and responsiveness to exercise for pre-diabetes treatment [62,63,64]. The present study used multiple analyses to classify participants into two groups, responders and non-responders to RBAC supplementation [65]. Four participants (P02, P03, P07, and P09) observed significant changes in all three classification criteria of individual beta diversity, weighted unifrac EMPeror plots, and taxonomic analysis, thus making them responders to RBAC supplementation. Two participants (P04 and P08) revealed changes in two criteria; hence, they were probable responders to RBAC supplementation. However, further supplementation and analysis is required to determine whether these participants can be classified as responders. Two participants (P01 and P10) only showed microbial taxonomic changes and therefore were unlikely to be responders to the supplementation. Further, two participants (P05 and P06) showing no statistical changes across the analyses were considered non-responders to the RBAC supplementation.

A limitation of this study is the small sample size. The number of recruited participants was constrained by the limitation of available resources (time and money). While the results demonstrated the potential modulation effect of RBAC supplementation on the gut microbiome, we caution against generalising the findings and drawing any definitive inference of the impact on the population. Further research with a larger sample size is required. As demonstrated in the present study, research aiming to investigate the effects of dietary RBAC supplementation on gut microbiota may require controlling for individual characteristics as confounding factors, such as age. Also, individualised doses according to body weight may be more appropriate to assess the impact of supplementation and dosage on gut microbiota. Future studies should also investigate the potential effects of RBAC supplementation on the gut microbiome at a molecular level, such as the impact on different SCFAs at the intestinal lumens. A further limitation of this study is the short time frame of the intervention. An extended period of supplementation may reveal further changes in the gut microbiota composition. Maintaining a food journal by participants to track the dietary intake and physical activity could be correlated with gut microbiota composition [50].

## 4. Materials and Methods

### 4.1. Participant Recruitment and Diet Tracking

The human research ethics committee at Charles Sturt University approved this research (protocol number ES03393) before the study commencement. Ten healthy adult volunteers, with an equal distribution of genders, were recruited through University presentations and social media posts. The inclusion criteria comprised generally healthy adults aged 18–60 years old with informed consent, and their health status at the time of joining was confirmed by a general practitioner. Exclusion criteria encompassed chronic health conditions and reported gut dysbiosis, ongoing medication use, history of antibiotic use in the three months prior to taking the first dietary supplement, and pregnancy or breastfeeding. Participants were required to maintain their usual diet with no major changes (e.g., changes from omnivore to vegan or vegetarian).

Baseline demographic data were collected in a screening survey before study commencement. Dietary intake was tracked by participants completing an online food frequency questionnaire, the AES, before study commencement and at study completion. Data collated from the AES detailed the participants’ macronutrient intake as well as the percentage of discretionary and core food intakes. An ARFS was obtained from the survey results for each participant [66]. These scores were categorised into very low (<23), low (24–29), medium (30–33), high (34–40), and very high (>40) levels of core food intakes in the diet.

### 4.2. Design and Intervention

The study intervention required participants to supplement their regular diet with two doses of RBAC, a low dose (1 g per day for 6 consecutive weeks) and a high dose (3 g per day for 6 consecutive weeks). RBAC is commercially available as a dietary supplement in Australia. The supplement sachets used in this study were supplied by BioMedica Nutraceuticals Pty Ltd. (Sydney, NSW, Australia). Each RBAC sachet contained 1 g of RBAC as the active ingredient plus microcrystalline cellulose (0.50 g), modified starch (0.26 g), dextrin (0.20 g), and tricalcium phosphate (0.04 g) as excipients. The net weight in each sachet amounted to 2 g. Participants were instructed to take 1 sachet per day for 6 weeks (low dose), then 3 sachets per day for 6 weeks (high dose), with a washout period of 3 weeks between these doses. Two baseline faecal samples were collected to establish the individual baseline, and one post-intervention faecal sample was collected to assess for ongoing effects. The implementation of this design resulted in a total of 8 time points for faecal sample collection over 21 weeks. This interventional design was implemented as per the schedule shown in Table 6.

### 4.3. Faecal Sample Collection, Preservation, and DNA Extraction

Faecal samples were collected by participants using a faecal swab collection and preservation system (cat. 45670-B, Norgen Biotek Corp., Thorold, ON, Canada). Nucleic acid within the collected swab remains stable for up to 2 years with DNA stored in the buffer at room temperature [67]. Participants were instructed to swab their faeces at the start of the sampling week as per the sample schedule and place the swab in the provided vial containing buffer. Swabs were either collected or posted from participants and stored in a −80 °C freezer. Faecal samples were subjected to DNA extraction using the microbiome DNA isolation kit (cat. 64100, Norgen Biotek Corp., Thorold, ON, Canada). For the lysate preparation, swabs were briefly vortexed before 500 μL of the sample faecal/buffer mixture was transferred to a DNAase-free microcentrifuge tube. Lysis buffers E and A were added as per product procedure, and samples were incubated in a 65 °C water bath for 10 min. Column binding and column washing were performed as per product procedure. DNA elution was completed twice to improve total yield.

### 4.4. Amplification and Sequencing of Extracted Microbial DNA

The concentration and purity of the extracted DNA were determined for each sample using the Nanodrop 2000 spectrophotometer (ThermoFisher Scientific, Waltham, MA, USA). DNA concentration and purity (260/280 ratio) were noted for sequencing submission. As per MyTaq Red Mix procedure, PCRs were performed on randomly selected extractions to ensure the extracted DNA was primarily of microbial origin and suitable for downstream analysis. Primers (0341-CCTACGGGNGGCWGCAG and 0785-GACTACHVGGGTATCTAATCC 16S rRNA target primers) were diluted to a 1:5 ratio [68]. A master mix was prepared, and PCR was conducted as per MyTaq Red Mix insert procedure and electrophoresis using E-Gel precast Agarose Electrophoresis System 1%. Ten (10) μL of sample and 20 μL of a 1 kb DNA ladder (Cat: D0428, Sigma Aldrich, St Louis, MO, USA) were used, and two wells (10 and 11) were loaded as a positive and negative control, respectively.

Thirty (30) μL of the extracted DNA samples were prepared in a round bottom 96 well microplate, sealed and transported to Ramaciotti Centre for Genomics (UNSW, Sydney, NSW, Australia). DNA amplification and sequencing were performed on the Illumina MiSeq platform, targeting the gene region 16S rRNA V3-V4 using primers 341-forward and 805-reverse.

### 4.5. Microbiota Profiling and Statistical Analysis

Microbiota profiling and statistical analysis were performed using the QIIME2 platform [69]. Data were imported to QIIME2 following the Casava 1.8 paired end demultiplexed fastq instructions. The DADA2 plugin was used to find amplicon sequences, remove chimeric sequences, and minimise noise created by spurious operational taxonomic units [70]. The trim and truncating parameters (from 5′–3′) were run with truncating-length-forward 285 and truncating-length-reverse 220, and used for data analysis due to sequencing quality cut-offs and sequencing depths.

Phylogenetic taxonomy was assigned to the sequencing data using VSEARCH with QIIME-compatible SILVA release 138 SSURef NR99 full-length region sequences and 138 SSURef NR99 full-length region taxonomy [71]. Taxonomic changes were determined using the analysis of the composition of microbiota with the bias correction (ANCOM) method, where the statistical significance was determined using a W value [72]. A one-way ANOVA was performed on the data to determine significant changes over time.

Alpha diversity (richness and evenness) and beta diversity (change in compositions over time) analyses were completed using QIIME2 q2-diversity plugin based on a phylogenetic diversity tree. This was generated with a sampling depth of 26,430 determined by 56.32% of retained features in 100% of samples. Alpha diversity applying Shannon’s evenness and Faith’s phylogenetic diversity (Faith’s PD) were calculated using the Kruskal–Wallis H test, with the data being pooled and ranked from smallest to largest (one-way ANOVA on ranks).

Beta diversity was calculated using four different methods: unweighted unique fraction metric (unifrac), weighted unifrac, Bray–Curtis, and Jaccard. Statistical analysis of beta diversity was based on the permutational multivariate analysis of variance (PERMANOVA). Analyses were performed to identify associations of beta diversity with explanatory factors and the RBAC supplementation. The Adonis PERMANOVA test for beta diversity significance in QIIME2 was used to determine significant factors that affect beta diversity through fitting polynomial regression models [73].

Participants were classified as responders to RBAC intervention if they exhibited significant changes across three analysis methods (with *p* < 0.05). This included individual beta diversity analysis, changes to beta diversity visualised with Earth Microbiome Project (EMPeror) plots across experimental phases, and alterations in taxonomic compositions. Multiple analyses were performed to reduce the likelihood of false positives due to the limited sample size.

## 5. Conclusions

Overall, the present study revealed that individual factors such as diet and lifestyle significantly influence alpha and beta diversity of human gut microbiota. Dietary supplementation of RBAC was shown to influence the beta diversity of gut bacteria of most participants. Based on individual changes in the gut microbiota profile, participants were classified as responders or non-responders. RBAC supplementation was shown to influence the abundance of the beneficial bacteria, *Bifidobacterium*, *Lactobacillus*, *Roseburia*, *Ruminococcaceae*, and *Faecalibacterium* in five participants. These findings suggest that the fermentation and degradation of RBAC by gut microbiota may be highly individualised. This evidence advocates further research controlling for participant characteristics, such as age and lifestyle, during dietary supplementation for the gut microbiota in healthy adults.

## Figures and Tables

**Figure 1 molecules-28-05400-f001:**
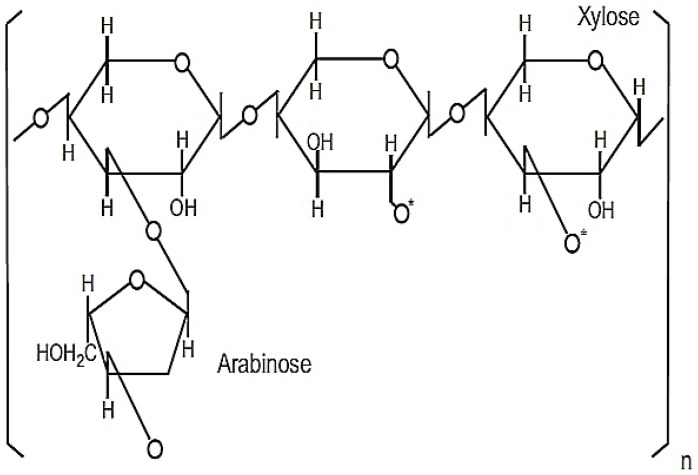
Chemical structure of MGN-3/Biobran. Reprinted from *Wheat and Rice in Disease Prevention and Health* [41], Copyright 2014, with permission from Elsevier.

**Figure 2 molecules-28-05400-f002:**
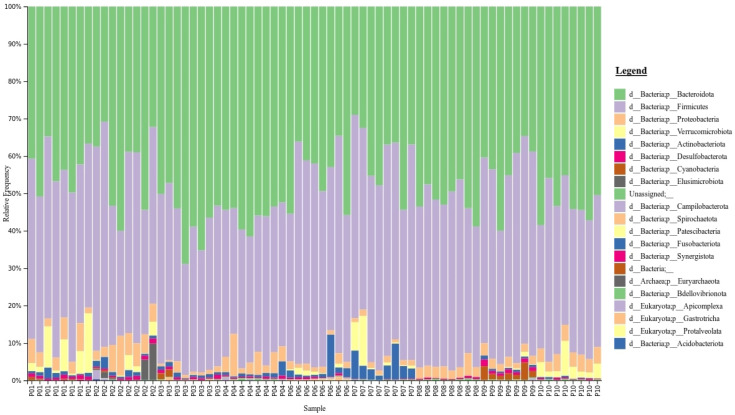
Taxonomy of the gut microbiota for participants across all sample time points visualised at the phylum level. Samples from each of the participants are represented on the *x*-axis with time points 1–8 within each section.

**Figure 3 molecules-28-05400-f003:**
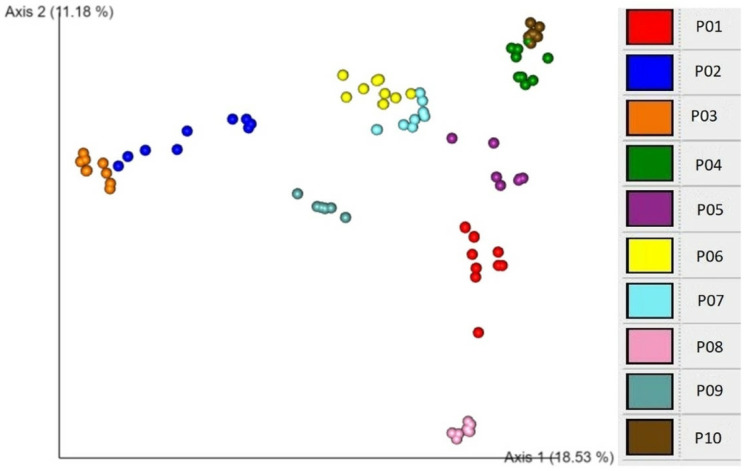
Bray–Curtis beta diversity EMPeror plot of all participants. Each point represents one sample from the corresponding participant and is coloured per participant for enhanced visualisation. Numbers on the axes refer to the percentage of variation explained by each axis of ordination.

**Figure 4 molecules-28-05400-f004:**
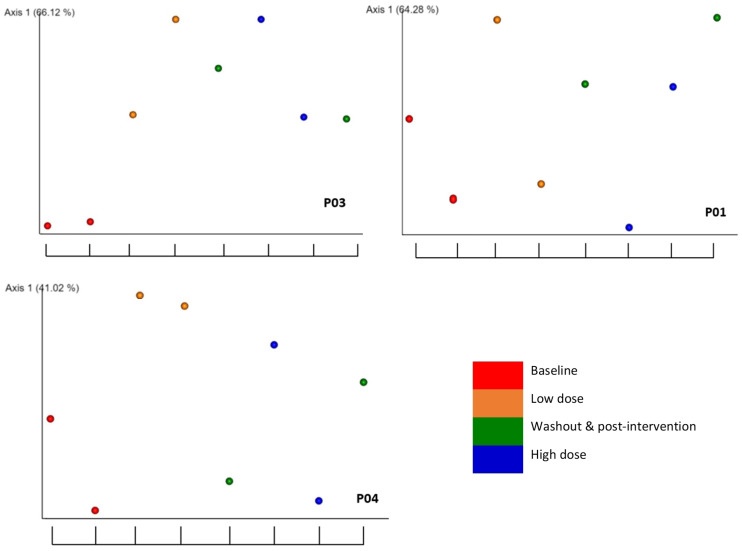
EMPeror plots of weighted unifrac beta diversity distance matrix among selected participants. Axis 1 indicates the percentage of variation of the total variance between samples (alteration of microbial composition) and *x*-axis indicates days since experimental phase began, depicting time points 1–8. Distance on *y*-axis indicates similarity of samples.

**Table 1 molecules-28-05400-t001:** Participants’ characteristics and dietary intake from the screening survey and Australian Eating Survey.

Participant	Sex	Age	Age Group	Cigarette Smoking	Alcohol Consumption	Diet	ARFS Average	ARFS Group ^1^
P01	F	26	≤30	No	No	Omnivore	16	Very low
P02	M	29	≤30	No	Yes	Omnivore	31.5	Medium
P03	M	25	≤30	No	No	Vegan	42.5	Very high
P04	F	26	≤30	No	Yes	Omnivore	29	Low
P05	F	27	≤30	No	Yes	Pescatarian	28	Low
P06	M	22	≤30	No	Yes	Omnivore	25.5	Low
P07	F	56	>30	Yes	No	Omnivore	32	Medium
P08	M	37	>30	No	Yes	Omnivore	34	High
P09	F	28	≤30	No	No	Omnivore	16.5	Very low
P10	M	30	≤30	No	Yes	Omnivore	22	Very low

^1^ ARFS group was classified into very low (<23), low (24–29), medium (30–33), high (34–40), and very high >40, according to ARFS average.

**Table 2 molecules-28-05400-t002:** Associations of alpha diversity measured with Shannon’s evenness and Faith’s PD for explanatory factors using Kruskal–Wallis analysis (*p* values below 0.05 are indicated in bold).

Explanatory Factor	Shannon’s Evenness	Faith’s PD
Participant	**9.66** × **10**${}^{-8}$	**0.002**
Alcohol consumption	**7.92** × **10**${}^{-6}$	0.389
Cigarette smoking	**0.008**	0.113
Sex	**0.003**	0.165
ARFS group	**0.032**	0.494
Age group	0.390	0.149
Interventional period ^1^	0.394	0.602
RBAC dosage ^2^	0.592	0.442
Experimental phase ^3^	0.756	0.442
Time point ^4^	0.875	0.527

^1^ Interventional period compares baseline with the combination of low dose, washout, high dose, and post-intervention periods. ^2^ RBAC dosage 0 g/day, 1 g/day, and 3 g/day. ^3^ Experimental phase is the comparison of baseline with low dose, washout, high dose, or post-intervention. ^4^ Time point includes 8 time points over 21 weeks.

**Table 3 molecules-28-05400-t003:** Associations of beta diversity with explanatory factors determined using PERMANOVA analysis (*p* values below 0.05 are indicated in bold).

Explanatory Factor	Unweighted Unifrac	Weighted Unifrac	Bray–Curtis	Jaccard
Participant	**0.001**	**0.001**	**0.001**	**0.001**
Sex	**0.001**	**0.001**	**0.001**	**0.001**
Alcohol consumption	**0.001**	**0.001**	**0.001**	**0.001**
ARFS group	**0.001**	**0.001**	**0.001**	**0.001**
Age group	**0.001**	**0.013**	**0.001**	**0.001**
Cigarette smoking	**0.001**	0.100	**0.001**	**0.001**
Interventional Period ^1^	0.826	0.275	0.866	0.928
RBAC dosage ^2^	0.936	0.644	0.997	0.999
Experimental phase ^3^	0.992	0.730	1.000	1.000
Time point ^4^	1.000	0.998	1.000	1.000

^1^ Interventional period compares baseline with the combination of low dose, washout, high dose, and post-intervention periods. ^2^ RBAC dosage 0 g/day, 1 g/day, and 3 g/day. ^3^ Experimental phase is the comparison of baseline with low dose, washout, high dose, or post-intervention. ^4^ Time point includes 8 time points, 3 weeks apart over 21 weeks.

**Table 4 molecules-28-05400-t004:** Associations of beta diversity with explanatory factors using weighted unifrac and Bray–Curtis analyses for each participant (*p* values below 0.05 are indicated in bold).

Participant	Weighted Unifrac			Bray–Curtis		
	RBAC Dosage ^1^	Experimental Phase ^2^	Interventional Period ^3^	RBAC Dosage ^1^	Experimental Phase ^2^	Interventional Period ^3^
P01	0.200	0.413	0.471	0.141	0.258	0.168
P02	0.095	0.060	**0.010**	0.064	0.067	**0.012**
P03	0.520	**0.034**	**0.052**	0.129	**0.002**	**0.028**
P04	**0.035**	0.146	0.894	**0.018**	**0.044**	0.114
P05	0.269	0.162	0.316	0.082	0.060	0.174
P06	0.948	0.763	0.924	0.775	0.631	0.787
P07	0.086	0.102	0.062	0.068	**0.025**	**0.025**
P08	**0.046**	0.094	0.569	**0.039**	**0.001**	**0.025**
P09	**0.015**	**0.008**	0.945	**0.039**	**0.021**	0.271
P10	0.961	0.926	0.174	0.832	0.087	0.165

^1^ RBAC dosage 0 g/day, 1 g/day, and 3 g/day. ^2^ Experimental phase is the comparison of baseline with low dose, washout, high dose, or post-intervention. ^3^ Interventional period compares baseline with the combination of low dose, washout, high dose, and post-intervention periods.

**Table 5 molecules-28-05400-t005:** Statistically significant taxonomic changes from baseline to interventional period determined using ANCOM. Data displayed only for participants showing a W ≥ 5.

Participant	Bacteria (Phylum, Class, Order, Family, Genus, and Species)	W	Associated Change
P01	Firmicutes; Clostridia; Lachnospirales; *Lachnospiraceae*; *Hungatella*	40	increase
	Proteobacteria; Gammaproteobacteria; Enterobacterales; *Enterobacteriaceae*; *Citrobacter*	12	increase (washout)
	Firmicutes; Bacilli; Erysipelotrichales; *Erysipelatoclostridiaceae*; *Erysipelatoclostridium*	8	increase (washout)
	Firmicutes; Clostridia; Lachnospirales; *Lachnospiraceae*; *unknown*	6	decrease
P02	Firmicutes; Clostridia; Peptostreptococcales-Tissierellales; *Peptostreptococcales-Tissierellales*; *Anaerococcus*	64	decrease
	Actinobacteriota; Actinobacteria; Corynebacteriales; Corynebacteriaceae; *Corynebacterium*; *unknown*	63	decrease
	Actinobacteriota; Actinobacteria; Corynebacteriales; Corynebacteriaceae; *Corynebacterium*; *unknown*	50	decrease
	Firmicutes; Clostridia; Peptostreptococcales-Tissierellales; *Peptostreptococcales-Tissierellales*; *Finegoldia*	48	decrease
P03	Negativicutes; Veillonellales-Selenomonadales; Veillonellaceae; *Dialister*	94	decrease
	Cyanobacteria; Vampirivibrionia; Gastranaerophilales; *Gastranaerophilales*; *Gastranaerophilales*	86	decrease
P07	Firmicutes; Bacilli; Erysipelotrichales; *Erysipelatoclostridiaceae*	102	decrease
P08	Firmicutes; Clostridia; Oscillospirales; *Ruminococcaceae*; *Eubacterium siraeum* group	35	decrease
	Firmicutes; Clostridia; Lachnospirales; *Lachnospiraceae*; *Eisenbergiella*	9	decrease
	Firmicutes; Clostridia; Lachnospirales; *Lachnospiraceae*; *Lachnospiraceae ND3007* group	7	increase
	Proteobacteria; Gammaproteobacteria; Enterobacterales; *Enterobacteriaceae*; *Escherichia-Shigella*	6	increase (low dose)
	Firmicutes; Clostridia; Lachnospirales; *Lachnospiraceae*; *Frisingicoccus*	5	increase
P09	Firmicutes; Clostridia; Oscillospirales; *Ruminococcaceae*; *Eubacterium siraeum* group	38	increase
	Firmicutes; Clostridia; Peptostreptococcales-Tissierellales; *Peptostreptococcales-Tissierellales*; *Anaerococcus*	8	increase (low dose)
	Firmicutes; Clostridia; Oscillospirales; *Ruminococcaceae*; *Ruminococcus*	8	increase (high dose)
	Firmicutes; Negativicutes; Veillonellales-Selenomonadales; *Veillonellaceae*; *Megasphaera*	8	increase (low dose)
	Firmicutes; Bacilli; Erysipelotrichales; *Erysipelotrichaceae*; *Solobacterium*	7	increase (low dose)
	Firmicutes; Clostridia; Lachnospirales; *Lachnospiraceae*; *Eubacterium hallii* group	5	increase
	Firmicutes; Bacilli; Erysipelotrichales; *Erysipelotrichaceae*; *Turicibacter*	5	increase (high dose)
	Firmicutes; Clostridia; Lachnospirales; *Lachnospiraceae*; *Stomatobaculum*	5	increase
P10	Bacteroidota; Bacteroidia; Bacteroidales; *Prevotellaceae*; *Prevotella*	20	decrease
	Firmicutes; Clostridia; Oscillospirales; *Butyricicoccaceae*; *Butyricicoccus*	8	increase
	Firmicutes; Bacilli; Lactobacillales; *Carnobacteriaceae*; *Granulicatella*	5	increase

**Table 6 molecules-28-05400-t006:** Dosage and schedule of RBAC supplementation with corresponding faecal sample number.

Time Point	RBAC Dose	Faecal Sample	Experimental
(Week)	(g/day)	Number	Phase
−3	0	1	Baseline
0	0	2	Baseline
3	1	3	Low dose
6	1	4	Low dose
9	0	5	Washout
12	3	6	High dose
15	3	7	High dose
18	0	8	Post-intervention

## Data Availability

The data that support the findings of this study are available from the corresponding authors, (S.W. or S.C.P.), upon reasonable request.

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
