# Peer review of "Changes in the Human Gut Microbiome during Dietary Supplementation with Modified Rice Bran Arabinoxylan Compound"

_molecules, 2023, doi:10.3390/molecules28145400_

Round 1

Reviewer 1 Report

This work aims to present the effect of modified rice bran arabinoxylan compound as a dietary supplement on the gut microbiota of healthy adults. However, the main concern is the number of individuals evaluated during the clinical study.

In order to obtain relevant results and have a robust statistical significance ten individuals are not enough. In addition, the recruitment criteria are not strict enough in order to achieve an homogeneous population. Furthermore, the health status is not supported by any means (clinical evaluation by an M.D., blood tests, any other check up).

The microbiota analysis offers interesting results and describes properly the different types of bacteria present in all the samples, however, this cannot be related to the clinical intervention due to a low n.

Table.5 is the evidence of the weakness of this work, since it is not acceptable a description of individual results in a work that aims to describe the effects on a population.

If the authors are able to increase the population to at least n=30, then the impact, significance and analysis would be improved and could be considered for publication.

Another possibility would be to change the aim of the study from a "clinical study" to a "basic science study" and increase the description at a molecular level of this ten patients relating the type of microbiota to other biochemical/immunological parameters

Author Response

We thank the reviewer for the valuable feedback. For this small study, the number of participants recruited was restricted by the limited resources available. We agree that this research is more appropriately designated as a basic science study rather than a clinical study. We have replaced the mention of “clinical study” with “basic science study” in the manuscript (lines 90, 210).

To clarify, we did ensure that only generally healthy individuals could join the study to achieve a more homogeneous cohort. We required the participants to obtain clearance from their GPs before entering the study (see lines 331-332).

Our manuscript has highlighted that the microbiome is highly individualised and not homogenous, with the variable “Participant” explaining approximately 80% (R2=0.796, p=0.001) of variance in the beta diversity using the Adonis PERMANOVA test. However, even with a small number of participants (n=10), we did detect a statistical significance for the variables “Experimental phase” (R2=0.019, p=0.026) and “Intervention” (R2=0.012, p=0.011), explaining approximately 3.1% of the variance observed in beta diversity (line 183-187). Thus, this suggests that the intervention with RBAC did have significant effects even on the non-homogenous population.

Responder analysis is a valid experimental outcome for a scientific study. Table 5 illustrates the individualised taxonomic changes necessary to determine the number of potential responders to the intervention. Statistically significant taxonomic changes (from baseline) were detected down to a species level for 7 participants, which is used for responder analysis. Hence, we are using the aggregation of individualised responses as the parameter for determining the responders and drawing conclusions for the group of participants.

Notwithstanding, we acknowledge that the small number of participants is a limitation of this study and caution against drawing any definite inference about the effects on the population. We have updated the manuscript to stress this point and suggested further research with a large sample size (lines 310-314). Furthermore, we recommend that future studies investigate the potential effects of RBAC supplementation on gut microbiome at a molecular level, such as the impact on different SCFAs at the intestinal lumens (lines 319 – 321).

Reviewer 2 Report

Title: Changes in the human gut microbiome during dietary supplementation with modified rice bran arabinoxylan compound 

Comment 1

I have some comments on the results presented in Figure 1. In this figure, it can be observed that some participants have 7, 8, or 9 time points. The authors in Figure 1 refer to technical replicates. It is not clear which time points are replicated in this figure. What do authors mean by replicate? Is it a replica of a particular time point analysis? The explanation about the replicas should be clear. The authors reported that three fecal samples were lost. Were they from different participants? This could be indicated in the text if each participant has a different number of time points.

Please clarify this information about the replicates and the different time points among the 10 participants. My suggestion is to not present the replicates in Figure 1, but instead present the average result of those replicates. However, the replicates are clearly presented in Figure 3, but as they were not applied to all samples, it is confusing why some time points have replicates.

Comment 2

Authors should compare the results (lines 117–119) with the conclusion (line 397). The conclusion that "age influences alpha diversity of human gut microbiota", is contradictory with the result presented in the results: "There were no significant changes in alpha diversity across age groups.". Please explain.

Line 82, complete the information indicating the average group (n=10) age (30,5 years).

Lines 153 and 154, text is repeated.

Line 179-190, authors should indicate the colours associated with these gut microbiota in Figure 1, to compare the changes described in the text.

Line 197, “explicated”, explained

Line 304, give information about gender distribution 5F:5M

Line 316, a bracket is missing

Table 1, correct ARFS group classification from “high” to “very high” in Line P03

Figure 1, indicate the colour correspondence to the taxonomy identified

Tables 2, 3 and 4, “(p values below 0.05 indicated in bold”, between brackets

Author Response

We greatly appreciate your valuable feedback. Technical replicates are repeated measurements of the same sample representing independent measures of the random noise associated with protocols or equipment. Only a small number of the samples from different participants are randomly selected for replicate testing. Hence only some time points are replicated. However, we agree that showing replicates in the figures can be confusing. As suggested, we have updated the manuscript by removing the replicates from the figures (Figures 2, 4, and S1) and all mentions of replicates in the texts. We have also reported which three fecal samples were lost in the text (see line 108).

Thank you for pointing out the discrepancy between the results and our conclusion on age influences on the gut microbiome. We have replaced age with diet instead (line 426). Dietary pattern based on the ARFS group is an explanatory factor significantly associated with the samples’ alpha diversity (Shannon) and beta diversity (All measures).

We have also updated the manuscript based on suggestions as follows:

Line 82, complete the information indicating the average group (n=10) age (30,5 years).

  • We have added (n=10, average age of 30.6 years) to the statement (Line 100 in the updated manuscript).

Lines 153 and 154, text is repeated.

  • The repeated text was deleted. See line 173 in the updated manuscript.

Line 179-190, authors should indicate the colours associated with these gut microbiota in Figure 1, to compare the changes described in the text.

  • We are unable to match the colours associated with the gut microbiota in Figure 1 (Figure 2 in the updated manuscript). The data shown in the figure is at Phylum levels, whereas the significant taxonomic changes presented at lines 197-208 (updated) are at the genus level and below. In any case, we have added a colour legend to Figure 2 for clarity.

Line 197, “explicated”, explained

  • We have updated the statement to “… could be affected by a couple of factors.” (line 215 updated) Then follow with the explanations of these factors.

Line 304, give information about gender distribution 5F:5M

  • We have added “… with an equal distribution of gender. “ (line 329 updated).

Line 316, a bracket is missing

  • Closing bracket added (line 343 updated).

Table 1, correct ARFS group classification from “high” to “very high” in Line P03

  •  

Figure 1, indicate the colour correspondence to the taxonomy identified.

  • A new legend is added to Figure 2 in the updated manuscript.

Tables 2, 3 and 4, “(p values below 0.05 indicated in bold”, between brackets 

  • All updated.

Reviewer 3 Report

This work investigated the effects of modified rice bran arabinoxylan compound (RBAC) as a dietary supplement on the gut microbiota of healthy adults.

Comments:

1. Details of the arabinoxylan molecules and their structure do not appear in all manuscripts, they must be provided by the author, including the PubChem CID and/or CAS Number.

2. Authors have stated "modified rice bran arabinoxylan compound (RBAC)", but details on the method: how to modify ice bran arabinoxylan compound? how is RBAC provisioned and isolated? using what method? it's still not clear.

3. The aspect of toxicity should be discussed in the introduction, as an introduction so that this manuscript appears to be more well organized.

4. Why are only 10 subjects used? what is the selection based on?

5. What is the RBAC dose given to the subject based on? it's still not clear.

6. Each figure and table must have a legend.

Grammar needs to be rechecked by authors

Author Response

Thank you for your valuable feedback. Please see our point-by-point responses below.

  1. Details of the arabinoxylan molecules and their structure do not appear in all manuscripts, they must be provided by the author, including the PubChem CID and/or CAS Number.

Authors: We have added a figure of RBAC’s suggested molecular structure (see Figure 1 in the updated manuscript). However, there is no PubChem CID and/or CAS Number available for this molecular structure.

  1. Authors have stated “modified rice bran arabinoxylan compound (RBAC)”, but details on the method: how to modify ice bran arabinoxylan compound? how is RBAC provisioned and isolated? using what method? it’s still not clear.

Authors: We have added further details on the production of RBAC (lines 70-76).

  1. The aspect of toxicity should be discussed in the introduction, as an introduction so that this manuscript appears to be more well organized.

Authors: We have added more details on the toxicity and safety of RBAC in the introduction section (lines 77-87).

  1. Why are only 10 subjects used? what is the selection based on?

Authors: We have explained that the number of recruited participants was constrained by the limited available resources (time and money) and acknowledged the small sample size as a study limitation (lines 311-314). This research is now described as a basic science study with explorative intent (lines 90-91, 210).

  1. What is the RBAC dose given to the subject based on? it's still not clear.

The dosage is typically used in RBAC studies and is generally considered safe. We have added this explanation in the introduction section (line 84).

  1. Each figure and table must have a legend.

We have ensured that all figures and tables have a legend. A new legend is added to Figure 2.

Reviewer 4 Report

1. First and foremost, a visualization of the structure or figure of "rice bran arabinoxylan compound" should be presented in the introduction. This is mandatory, given that this manuscript was submitted and is under consideration in the "Molecules Journal".

2. In the method, provide clear details, how the "rice bran arabinoxylan compound" was isolated, obtained and processed to be added to the voluntary diet.

3. Mention the Ethical Approval on the method, even though it has been mentioned at the end of the manuscript, but it needs to be stated in the method section.

4. Why only 10 subjects/volunteers? How is sample size determined? What sample size formula do you use and how? Editors and authors must clearly see this concern.

5. "Dosage and schedule of RBAC supplementation" will be more interesting if it is statistically processed against the microbiota profile, using Pearson Correlation or Spearman; see example in figure 5 in publication: https://doi.org/10.3390/nu15040909. Believe me, this will cause the manuscript to be more rigorous and increase readability.

Author Response

Thank you for your valuable feedback. Please see our point-by-point responses below.

  1. First and foremost, a visualization of the structure or figure of "rice bran arabinoxylan compound" should be presented in the introduction. This is mandatory, given that this manuscript was submitted and is under consideration in the "Molecules Journal".

Authors: We have added a figure of RBAC’s suggested molecular structure (see Figure 1 in the updated manuscript). 

  1. In the method, provide clear details, how the "rice bran arabinoxylan compound" was isolated, obtained and processed to be added to the voluntary diet.

Authors: We have more details on the production of RBAC in the introduction (lines 70-76) and specify that the RBAC used in this study is available commercially as a dietary supplement and the company supplying the RBAC (lines 348-350) in the methods section. 

  1. Mention the Ethical Approval on the method, even though it has been mentioned at the end of the manuscript, but it needs to be stated in the method section.

Authors: We have added the Ethical Approval in the methods section (lines 327-328).

  1. Why only 10 subjects/volunteers? How is sample size determined? What sample size formula do you use and how? Editors and authors must clearly see this concern.

Authors: We have explained that the number of recruited participants was constrained by the limited available resources (time and money) and acknowledged the small sample size as a study limitation (lines 311-314). This research is now described as a basic science study with explorative intent (lines 90-91, 210).

  1. "Dosage and schedule of RBAC supplementation" will be more interesting if it is statistically processed against the microbiota profile, using Pearson Correlation or Spearman; see example in figure 5 in publication: https://doi.org/10.3390/nu15040909. Believe me, this will cause the manuscript to be more rigorous and increase readability.

Authors: Thank you for the suggestion. We can consider this when a secondary study is performed.

Round 2

Reviewer 1 Report

Thank you for the responses

Reviewer 3 Report

The revision is great